# Phytochemical Profiles and Cellular Antioxidant Activities in Chestnut (*Castanea mollissima* BL.) Kernels of Five Different Cultivars

**DOI:** 10.3390/molecules25010178

**Published:** 2020-01-01

**Authors:** Xiaoxiao Chang, Fengyuan Liu, Zhixiong Lin, Jishui Qiu, Cheng Peng, Yusheng Lu, Xinbo Guo

**Affiliations:** 1Institute of Fruit Tree Research, Guangdong Academy of Agricultural Sciences; Key Laboratory of South Subtropical Fruit Biology and Genetics Resource Utilization, Ministry of Agriculture (MOA); Guangdong Province Key Laboratory of Tropical and Subtropical Fruit Tree Research, Guangzhou 510640, China; xxchang6@163.com (X.C.); lzxf200@126.com (Z.L.); RGXOII307@126.com (J.Q.); pengcheng2007@foxmail.com (C.P.); 2School of Food Science and Engineering, South China University of Technology, Guangzhou 510640, China; phoyueen@outlook.com

**Keywords:** chestnut, phenolics, flavonoids, antioxidant activity

## Abstract

In this study, the phytochemical profiles, total and cellular antioxidant activities of five different Chinese chestnut (*Castanea mollissima* BL.) cultivars were analyzed. Phenolics, flavonoids as well as phytochemical compounds in five cultivars of chestnut kernels were determined. Results showed that the free forms played a dominant role in total phenolics, flavonoids and antioxidant activities of all five cultivars of chestnut kernels. The cultivar ‘Fyou’ showed the highest total and free phenolic contents, ‘Heguoyihao’ showed the highest total and free flavonoids contents, and ‘Chushuhong’ showed the highest total and cellular antioxidant activities. Eight phenolic compounds were detected, and chlorogenic acid, gallic acid, and quercetin were shown as three predominant components in all five cultivars. These results provide valuable information which may be a guidance for selection of good chestnut variety to be used as functional food.

## 1. Introduction

Chestnut is a crop consumed mainly in East Asia, Europe, and North America. *Castanea mollissima* (China), *Castanea Americana* (North America), *Castanea sativa* (European), and *Castanea crenata* (Korea and Japan) are four important species of chestnut [1,2,3]. Chestnut kernels contain rich carbohydrates and protein but zero gluten and cholesterol, and they are healthy source of stable food. It has been demonstrated that chestnut is also rich in vitamin E and carotenoids [2,4]. In addition, chestnut fruit has long been considered beneficial to the functions for human as Chinese traditional medicine in *“Compendium of Materia Medica”* [5], and it has also been used as cosmetic supplement in Korea peninsula [6].

Epidemiological studies have demonstrated the significance of phytochemicals, especially phenolic compounds, to human health due to their additive and synergistic effects on radicals scavenging [7,8,9]. Recent years, more studies focused on phytochemical profiles and antioxidant activities of fruits [10,11,12,13,14,15], and also the nuts [16,17], such as walnut [18,19], pistachio [20,21], and chestnut [1,3,22,23,24,25].

Like other plant species, the varieties, even the growth condition can significantly affect the phytochemical concentration in chestnut fruits [26,27]. Otles and Selek [3] have conducted a thorough study about the phenolic compounds and antioxidant activities of chestnut fruits collected from 16 provinces in Turkey. They discovered huge variation of total phenolic contents among chestnut fruits from different provinces, but no obvious differences in terms of total antioxidant in vitro. However, little is known about the bioactivity, especially cellular antioxidant activities of these compounds in chestnut fruits. Furthermore, most studies on chestnut mainly focused on free phytochemicals, while bound phytochemicals and its contributions to the total phenolics and total antioxidant activity were overlooked [16,22,23,28].

In this study, the free and bound phenolic contents, flavonoid contents as well as phytochemical profiles in five different cultivars of chestnut kernels (Mifengqiu (MF), Dahongpao (DH), Chushuhong (CS), Heguoyihao (HG), and Fyou (FY)) were determined. In addition, total and cellular antioxidant activities were evaluated by oxygen radical absorbance capacity (ORAC) assay and cellular antioxidant activity (CAA) assay, respectively. The intention of present work is to make a comprehensive comparison of these five cultivars of chestnuts to provide valuable information for high-quality breeding selection and functional food innovation in chestnut.

## 2. Results

### 2.1. Phenolic Contents of Five Different Chestnut Cultivars

The total phenolic contents including free and bound phenolics of five different chestnut cultivars were evaluated and results were shown in Figure 1. The total phenolic contents of the five different cultivars varied from 42.8 to 58.6 mg GAE/100 g FW (GAE, gallic acid equivalent). The cultivar ‘FY’ showed the highest total phenolic content of 58.60 ± 4.2 mg GAE/100 g FW, and the highest free phenolics of 40.25 ± 3.5 mg GAE/100 g FW, which was followed by cultivar ‘HG’ with 52.95 ± 0.4 mg GAE/100 g FW of total phenolics and 33.14 ± 0.9 mg GAE/100 g FW of free phenolics. The lowest one was ‘MF’, with total and free phenolics contents of 42.77 ± 1.2 mg GAE/100 g FW, and 24.35 ± 0.9 mg GAE/100 g FW respectively. The bound phenolics of the five different cultivars varied from 15.4 to 19.8 mg GAE/100 g FW, and the values was not correspondence with the levels of total and free phenolics. In terms of bound phenolics, the highest one was ‘HG’, not ‘FY’; and the lowest one was ‘DH’, not ‘MF’. According to the results, the total phenolics of chestnut kernel mainly existed as free form. The ratio of free phenolics compared to total phenolics varied from 57% to 69% in the five different cultivars.

### 2.2. Flavonoid Contents of Five Different Chestnut Cultivars

The total flavonoid contents including free and bound flavonoids of five different chestnut cultivars were evaluated and results were shown in Figure 2. The total flavonoid contents varied from 122 to 218 mg CE/100 g FW (CE, catechin equivalent) in five different cultivars. ‘HG’ and ‘MF’ all showed the highest values of total and free flavonoids, with 217.6 ± 9.7 and 168.5 ± 3.0 mg CE/100 g FW, 206 ± 18.3 and 155.2 ± 18.2 mg CE/100 g FW respectively. This was followed by cultivar ‘FY’, and then ‘CS’ and ‘DH’. The bound flavonoids in five different varieties did not show significant differences between each other, and were around 50 mg CE/100 g FW. Results suggested that the total flavonoids in chestnut kernel mainly existed as free form, which contributed up to 77% to total flavonoids.

### 2.3. Phenolic Profiles of Five Different Chestnut Cultivars

As shown in Table 1, eight phenolic compounds were detected in chestnut kernel, including ferulic acid, chlorogenic acid, gallic acid, vanallic acid, syringate, 7-hydroxycoumarin, quercetin, and quercetin 3-rhamnoside, in which 7-hydroxycoumarin was only detected in bound form. The content of ferulic acid was the lowest in chestnut kernel than other compounds. Free ferulic acid only existed in cultivar ‘DH’, with the content of 0.16 ± 0.01 mg/100 g FW, but was not detected in other four cultivars. The bound ferulic acid was detected in all five cultivars, with the content around 0.4 mg/100 g FW, and showed no significant differences between each other.

Chlorogenic acid and gallic acid were the two major compounds of phenolics in chestnut kernel, but the main form they presented was different. Chlorogenic acid mainly existed as bound form, with the content around 11 mg/100g FW, and the content of free form was around 4.7 mg/100 g FW, with no significant differences among cultivars. While the gallic acid was just the opposite existing mainly in free form with the content around 13 mg/100 g FW, and the bound form was around 2.3 mg/100 g FW, with the highest value of 3.01 ± 0.7 mg/100 g FW in ‘FY’ and the lowest value of 1.64 ± 0.03 mg/100 g FW in ‘DH’. Followed by quercetin, which mainly exited in bound form with the content around 6.1 mg/100 g FW, and the free form was around 1.8 mg/100 g FW with no significant differences among cultivars.

The contents of vanallic acid and syringate were less than chlorogenic acid, gallic acid, and quercetin. The free vanallic acid content was more than 2 mg/100 g FW in five different chestnut cultivars, and the highest one was 2.28 ± 0.02 mg/100 g FW in ‘FY’. The bound vanallic acid content was less than 1.39 mg/100 g FW in different cultivars, and there were no significant differences between each other. The syringate in the chestnut kernel existed almost half as free form and half as bound form, and the contents in different cultivars were around 1.6 mg/100 g FW. The cultivar ‘HG’ had the highest content of free syringate with the value of 1.73 ± 0.01 mg/100 g FW, while the highest content of bound syringate was 1.77 ± 0.16 mg/100 g FW in ‘CS’. Quercetin 3-rhamnoside was detected in both free and bound form in cultivar ‘DH’ and ‘CS’, while it only existed in bound form in other three cultivars. The cultivar ‘DH’ showed the highest contents of both free and bound quercetin 3-rhamnoside than the others, with the value of 1.83 ± 0.03 mg/100 g FW and 3.66 ± 0.88 mg/100 g FW respectively. 7-hydroxycoumarin was detected only in bound form in five different cultivars, and the highest content was 1.96 ± 0.18 mg/100 g FW in ‘MF’ and 1.95 ± 0.09 mg/100 g FW in ‘CS’.

### 2.4. Total Antioxidant Activities of Five Different Chestnut Cultivars

The total antioxidant activities of five different chestnut cultivars were evaluated by ORAC assay. As shown in Figure 3, the highest total antioxidant activity was found in cultivar ‘CS’ with the value of 13.62 ± 1.4 μmol TE/g FW (TE, Trolox equivalent), which was followed by ‘HG’ and ‘FY’, with the values of 12.61 ± 1.5 μmol TE/g FW and 11.49 ± 1.8 μmol TE/g FW respectively. ‘MF’ and ‘DH’ showed the lowest value of 9.42 ± 0.6 and 9.23 ± 0.8 μmol TE/g FW respectively. The ORAC values of free phytochemicals in five different chestnut cultivars varied from 6.53 ± 0.6 to 10.23 ± 1.3 μmol TE/g FW, but there were no significant differences among each other. While the bound phytochemicals showed less ORAC values than free ones, with the highest value of 3.66 ± 0.3 μmol TE/g FW in ‘FY’ and the lowest value of 1.71 ± 0.1 μmol TE/g FW in ‘DH’. The contribution of free phytochemicals to total ORAC value varied from 68% to 81%, which suggested that free phytochemicals played the dominant role for the total antioxidant activity of chestnut kernel.

### 2.5. Cellular Antioxidant Activities of Five Different Chestnut Cultivars

Cellular antioxidant activities (CAA) of five different chestnut cultivars were evaluated in this study by CAA assay. PBS and No-PBS wash treatments were conducted in CAA assay to evaluate intracellular and extracellular antioxidant activities of phytochemical extracts, and also to analyze the cellular uptake of extracts. As shown in Table 2, the CAA values of free fractions were much higher than bound fractions, and in No-PBS wash samples, the values of free fraction were about 6–11 times higher than the ones in bound fractions. In PBS wash samples, the cultivars of ‘CS’, ‘HG’ and ‘FY’ were detected having the cellular antioxidant activities in free fractions, while others not. However, in bound fraction, just the cultivar ‘CS’ was detected having the cellular antioxidant activity. Comparatively, the cellular antioxidant activity of ‘CS’ showed the best performance than other cultivars both in No-PBS wash and PBS wash samples, and also in free or bound fractions, which showed the highest cellular uptake rates.

### 2.6. Correlation Analysis among Phenolics and Antioxidant Activities

The correlation of total phenolics, total flavonoids, and phenolic compounds with the total and cellular antioxidant activities were analyzed, and results were shown in Table 3. The content of total phenolics showed significant positive correlations with ORAC and CAA values (both PBS wash and no wash), especially No-PBS wash CAA (r = 0.71, *p* < 0.01). However, the total flavonoids did not show significant correlation with total antioxidant activities, neither ORAC nor CAA. For phenolic compounds, the total ferulic acid showed significant negative correlation with TPC (r = −0.66, *p* < 0.01) and PBS wash CAA (r = −0.59, *p* < 0.05). The total vanallic acid showed significant positive correlation with TPC (r = 0.80, *p* < 0.01) and No-PBS wash CAA (r = 0.63, *p* < 0.05). The total syringate and total quercetin both showed significant positive correlation with ORAC value. The total quercetin 3-rhamnoside showed significant negative correlation with both TPC (r = −0.57, *p* < 0.05) and TFC (r = −0.81, *p* < 0.01). Results suggested that the phenolics should be the most important parameter to evaluate the total antioxidant activity of chestnut kernel.

## 3. Discussion

### 3.1. Phenolics in Chestnut Were Underestimated

Phenolics are secondary metabolites in plants playing an important role in preventing cardiovascular diseases, cancers as well as diabetes [29]. Fruits like berries [30] are rich sources of phenolics, and the average total phenolic content of common fruits was around 200 mg/100 g FW [31]. Previous reports stated that the total phenolic contents in chestnut kernel were around 0.1–30 mg GAE/g DW [3,4,22,32].

Nevertheless, we believe that the content of polyphenols in chestnut kernel might be underestimated. On one hand, by Folin assay, we can only estimate the antioxidant content rather than the total polyphenols content, unless we apply this method to purified polyphenol samples [33]. On the other hand, to our knowledge, most studies on chestnut mainly focused on free phytochemicals, while bound phytochemicals and its contributions to the total phenolics and total antioxidant activity were overlooked [16,22,23,28]. As detected both free and bound phenolics by our research team, the total phenolics in chestnut kernel (fresh or treated) was found around 40–90 mg GAE/100 g FW and the value was much higher after steaming treatment than fresh chestnut kernel [2,22,24]. In this study, the contents of total phenolics varied from 40 to 60 mg GAE/100 g FW in five different chestnut cultivars, and the bound phenolics contributed about 31–43% to total phenolics (Figure 1), which might be overlooked in previous studies.

### 3.2. Phytochemical Profiles of Chestnut Kernel

Eight phenolic compounds were detected by HPLC from the five different chestnut cultivars (Table 1). In this study, the free and bound forms of these compounds were analyzed. Chlorogenic acid, gallic acid and quercetin presented as three predominant components, which exited in both free and bound forms in all five cultivars. In previous study, gallic and ellagic acid were payed much attention in European chestnut (*Castanea sativa* Mill.) grown in Portugal [1,26]. While ten phenolic compounds were detected from sixteen different chestnut fruits in Turkey, but only four (gallic acid, vanillic acid, rutin, and catechin) exited in at least fifteen cultivars, and vanillic acid was the most prominent compounds, quercetin was not found in any of these chestnuts studied [3]. However, rutin and catechin was not detected in our study about the five Chinese chestnut cultivars (*Castanea mollissima* BL.). This discrepancy might be caused by the differences between Chinese chestnut and European chestnut, and this hypothesis needs further research to demonstrate.

### 3.3. Antioxidant Activities of Chestnut Kernel

The total and cellular antioxidant activities of five different chestnut kernels were conducted by ORAC and CAA assays. The performance of the five different chestnut cultivars was similar in two different assays. The ‘CS’ cultivar showed the highest antioxidant activities analyzed by both ORAC and CAA assays, while ‘MF’ and ‘DH’ exhibited low antioxidant ability in these assays. The correlation analysis showed that the content of total phenolic positively correlated with ORAC and CAA values. This result was correspondence with previous studies, such as the study of cooking effect on antioxidant activities of four different Chinese chestnut cultivars showed a strong positive correlation between antioxidant activity and total phenolic content [32]. The study on antioxidant activity of pistachio, cashew and chestnut flours also showed total phenolics positively correlated with ORAC values [34]. The phenolics may contribute much important role in antioxidant activities of chestnut fruit, as studied previously in different components (leaf, flower, skin, fruit) of chestnut [22].

## 4. Materials and Methods

### 4.1. Materials Preparation

The chestnut cultivars: FY (Fyou), HG (Heguoyihao), CS (Chushuhong), DH (Dahongpao) and MF (Mifengqiu) were grown in deciduous fruits germplasm resource nursery in Institute of fruit tree research, Guangdong Academy of Agricultural Sciences (longitude 113.37 and dimension 23.15, Guangzhou, China). The chestnut fruits of each cultivar were collected at mature stage from three individual trees. Thirty fruits of each cultivar were selected for the following analysis, and every 10 fruits served as one replicate. The chestnut shell and inner skin were removed by hands and then the kernels were treated with nitrogen and ground into flour which was subsequently stored in 50 mL centrifuge tubes under −20 °C until analyzing.

### 4.2. Phytochemical Extraction

The extraction of phytochemicals was conducted using the protocol developed in our lab [35]. For free extracts, the chestnut kernels were grinded to flour with liquid nitrogen and immersed in 80% chilling acetone, and then treated by Bioruptor^®^ sonication system (Diagenode, Bioruptor. Co., Belgium) for 60 seconds before centrifugation. The centrifugation condition was 8000 rpm for 10 minutes at 4 °C. The liquid supernatant was collected by triplicates, and condensed by a rotary evaporator and reconstituted with 70% methanol. For bound extracts, the residues from the previous steps were collected and extracted by ethyl acetate and then reconstituted with 70% methanol. All the phytochemical extracts were stored at −20 °C until analyzing.

### 4.3. Determination of Phenolic and Flavonoid Contents

Folin–Ciocalteu assay was applied in this study to determine the phenolic contents of chestnut kernels according to previous study [35]. Gallic acid was used for calibration. Phenolic content was expressed as milligrams of gallic acid equivalent per 100 gram in fresh weight (mg GAE/100 g FW). Sodium borohydride/chloranil (SBC) assay was applied in this study to determine the flavonoid contents of chestnut kernels according to previous study [35]. Catechin was used for calibration. Flavonoid content was expressed as milligrams of catechin equivalent per 100 gram in fresh weight (mg CE/100 g FW).

### 4.4. Determination of Phytochemical Profiles

High-performance liquid chromatography (HPLC) with photodiode array detector was applied in this study for phytochemical composition analyzing according to previous study [25]. Mobile phase A: milli Q water with 0.1% trifluoroacetic acid; mobile phase B: chromatographically pure methanol; column: Waters Sun FireTM C18 column (250 mm × 4.6 mm, 5 μm); column temperature: 30 °C; flow rate: 1.0 mL/min. Gradient elution began with 3% eluent B, then 5% at 8 min, 10%B ate 15 min, 20%B at 25 min, 35%B at 52 min and finally 5%B at 60 min. UV absorption wavelengths were set at 280 nm and 324 nm to detect phenolics and flavonoids respectively. The qualification of phenolic compounds was obtained by comparison of retention time and recovery rate, while the quantification was carried out the calibration of external standard curves. Data were reported as milligrams per 100 gram in fresh weight (mg/100 g FW).

### 4.5. Evaluation of Total Antioxidant Activity

Oxygen radical absorbance capacity (ORAC) assay was applied to this study to determine the total antioxidant ability of chestnut kernels according to previous study [25]. Fluorescein was used as fluorescence probe and 2,2’-Aazobis dihydrochloride (ABAP) was used as free radical donor in this study. The dynamic fluorescence intensity of each sample was detected by a microplate reader (Thermal, Waltham, MA, USA). Total antioxidant activity was calculated with the standard Trolox and expressed micromole of Trolox equivalent per gram in fresh weight (µmol TE/g FW).

### 4.6. Evaluation of Cellular Antioxidant Activity

Cellular antioxidant activity (CAA) assay [36] was applied in this study to determine the cellular antioxidant ability of chestnut kernel extracts. Human live cancer cell line (HepG2, ATCC HB-8065) was used as cellular model in this assay; quercetin was used as standard to calculate the cellular antioxidant activity value. HepG2 cells were seeded at a density of 6 × 10^4^ cells/well on a 96-well microplate for antioxidant activity analysis. Dichlorofluorescin diacetate (DCFH-DA) was used as fluorescence probe and ABAP was used as free radical donor. With and without PBS wash treatments were used in this assay. Fluorescence intensity was measured at excitation of 485 nm and emission of 535 nm for a dynamic fluorescein intensity analysis by Multi-mode microplate reader (Molecular Devices, Sunnyvale, CA, USA). CAA value was calculated from the integrated area under the fluorescence versus time curve, and the results were expressed as nanomole of quercetin equivalents (QE) per gram in fresh weight (nmol QE/g FW).

### 4.7. Statistics Analysis

All the experiments were run three replicates and the data were reported as mean ± SD (n = 3). Figure plotting was achieved by Sigmaplot 11.0 (Systat Software, Inc, Chicago, IL, USA). Statistics analysis including significance analysis and Pearson correlation was calculated by SPSS 13.0 (SPASS Inc, Chicago, IL, USA).

## 5. Conclusions

This study analyzed both free and bound forms of phenolics, flavonoids and antioxidant activities of chestnut kernels. The free forms played dominant role in total phenolic, flavonoid, and antioxidant activities of all five different chestnut fruits. The cultivar ‘FY’ showed the highest total and free phenolic contents, ‘HG’ showed the highest total and free flavonoids contents, and ‘CS’ showed the highest antioxidant activities in both ORAC and CAA assays. Chlorogenic acid, gallic acid, and quercetin were shown as three predominant components existed in both free and bound forms in all five cultivars. These results provide valuable information of the phenolics and antioxidant activities of different chestnut kernels, and may be a good guidance for selection of better chestnut variety which will be used as functional food.

## Figures and Tables

**Figure 1 molecules-25-00178-f001:**
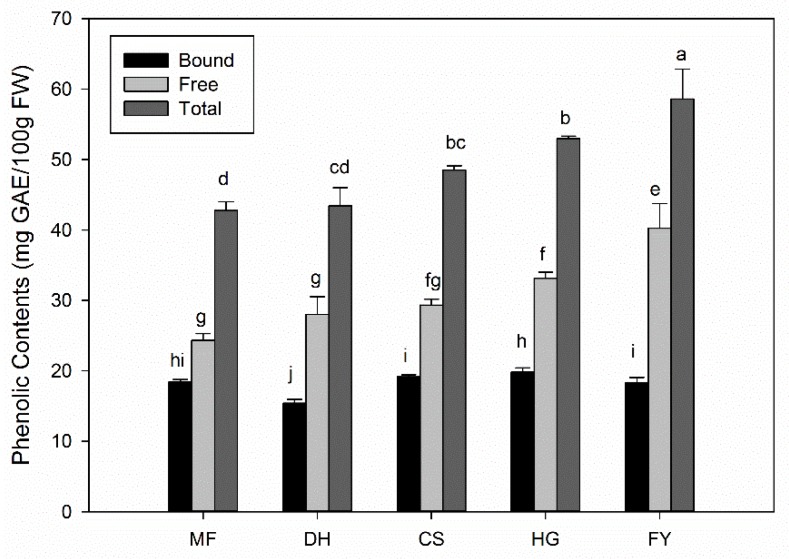
Phenolic contents of five different chestnut cultivars. Tukey’s test was carried out in each group (bound, free, and total) respectively and significant differences (*p* < 0.05) exist among those bars with different letters. MF—Mifengqiu; DH—Dahongpao; CS—Chushuhong; HG—Heguoyihao; FY—Fyou.

**Figure 2 molecules-25-00178-f002:**
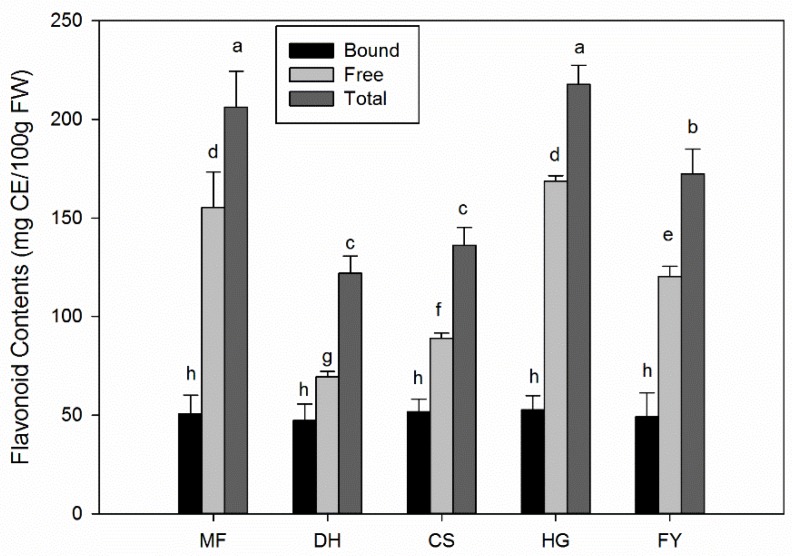
Flavonoid contents of five different chestnut cultivars. Turkey test was carried out in each group (bound, free and total) respectively and significant differences (*p* < 0.05) exist among those bars with different letters.

**Figure 3 molecules-25-00178-f003:**
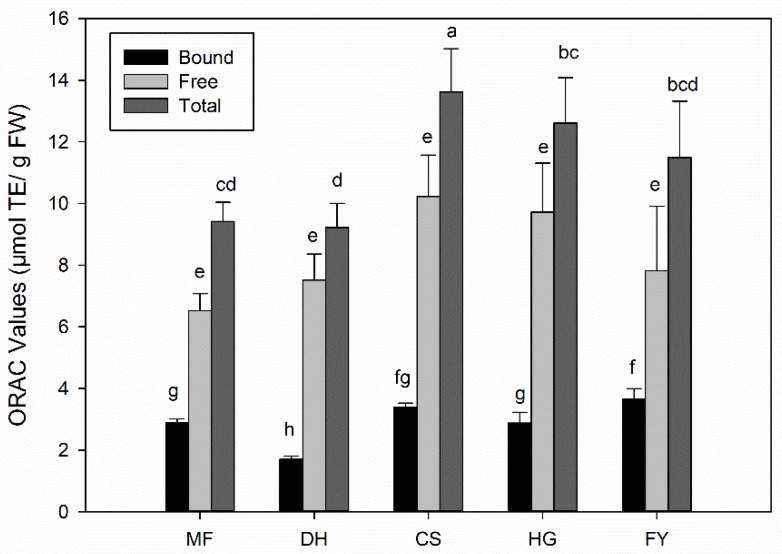
Total antioxidant activities of five different chestnut cultivars by ORAC assay. Turkey test was carried out in each group (bound, free and total) respectively and significant differences (*p* < 0.05) exist among those bars with different letters.

**Table 1 molecules-25-00178-t001:** Phytochemical profiles of five different chestnut cultivars.

Components	Conjugated Way	MF	DH	CS	HG	FY
Ferulic acid	free	ND	0.16 ± 0.01a	ND	ND	ND
	bound	0.44 ± 0.08a	0.41 ± 0.04a	0.40 ± 0.10a	0.40 ± 0.06a	0.32 ± 0.07a
Chlorogenic acid	free	4.71 ± 0.01a	4.72 ± 0.01a	4.75 ± 0.01a	4.75 ± 0.01a	4.69 ± 0.01a
	bound	11.18 ± 0.25a	12.22 ± 1.83a	11.12 ± 1.83a	11.25 ± 0.84a	9.75 ± 1.40a
Gallic acid	free	11.30 ± 0.09a	13.29 ± 1.58a	10.79 ± 2.41a	13.56 ± 0.80a	13.35 ± 2.93a
	bound	2.38 ± 0.79ab	1.64 ± 0.03b	2.59 ± 0.22ab	1.89 ± 0.57ab	3.01 ± 0.70a
Vanallic acid	free	2.07 ± 0.04b	2.09 ± 0.06b	2.08 ± 0.15b	2.13 ± 0.06ab	2.28 ± 0.02a
	bound	1.08 ± 0.03a	1.14 ± 0.13a	1.10 ± 0.02a	1.39 ± 0.13a	1.39 ± 0.23a
Syringate	free	1.69 ± 0.01c	1.70 ± 0.01b	1.63 ± 0.01d	1.73 ± 0.01a	1.64 ± 0.01d
	bound	1.49 ± 0.01ab	1.15 ± 0.30b	1.77 ± 0.16a	1.66 ± 0.03a	1.57 ± 0.08a
7-hydroxycoumarin	free	ND	ND	ND	ND	ND
	bound	1.96 ± 0.18a	1.80 ± 0.17b	1.95 ± 0.09a	1.73 ± 0.15c	1.79 ± 0.05b
Quercetin	free	1.84 ± 0.01a	1.84 ± 0.01a	1.85 ± 0.01a	1.84 ± 0.01a	1.85 ± 0.01a
	bound	5.25 ± 0.52a	6.14 ± 0.72a	7.05 ± 1.67a	6.60 ± 0.21a	5.80 ± 0.09a
quercetin 3-rhamnoside	free	ND	1.83 ± 0.03a	1.77 ± 0.01b	ND	ND
	bound	2.36 ± 0.34b	3.66 ± 0.88a	2.33 ± 0.21b	2.15 ± 0.18b	1.89± 0.05b

Notes: Unit, mg/100 g FW. Turkey tests were carried out in each row and significant differences (*p* < 0.05) exist among those with different letters. ND means not detected

**Table 2 molecules-25-00178-t002:** Cellular antioxidant activities of five different chestnut cultivars.

Cultivars	CAA Value (Nmol QE/g FW)				Cellular Uptake (%)	
	No PBS Wash		PBS Wash			
	Free	Bound	Free	Bound	Free	Bound
MF	1.39 ± 0.22b	0.20 ± 0.01ab	ND	ND	ND	ND
DH	2.22 ± 0.22a	0.18 ± 0.03b	ND	ND	ND	ND
CS	2.40 ± 0.03a	0.33 ± 0.08a	1.54 ± 0.11a	0.47 ± 0.01a	64.17	142.42
HG	2.69 ± 0.28a	0.29 ± 0.05ab	0.59 ± 0.09b	ND	21.92	ND
FY	2.72 ± 0.23a	0.33 ± 0.07a	1.62 ± 0.07a	ND	59.56	ND

Note: Tukey’s tests were carried out in each column and significant differences (*p* < 0.05) exist among those with different letters. ND means not detected. Cellular uptake (%) = CAA values of PBS wash/CAA values of no PBS wash.

**Table 3 molecules-25-00178-t003:** Pearson correlation coefficient among phenolics, flavonoids, phytochemical composition and antioxidant activities.

Correlation	TORAC	TCAA-No	TCAA-Wash	TPC	TFC
TORAC	−	0.54*	0.69 **	0.54 *	0.05
TCAA-No	−	−	0.61 *	0.71 **	−0.097
TCAA-wash	−	−	−	0.61 *	−0.241
TPC	−	−	−	−	0.203
TFC	−	−	^−^	−	−
TFA	−0.46	−0.36	−0.59 *	−0.66 **	−0.31
TCA	0.02	−0.14	−0.34	−0.37	−0.12
TGA	0.20	0.37	0.12	0.57 *	0.09
TVA	0.21	0.63*	0.28	0.80 **	0.30
TS	0.60 *	0.36	0.50	0.31	0.36
T7H	0.04	−0.33	0.07	−0.41	−0.02
TQ	0.57 *	0.49	0.40	0.13	−0.23
TQ3	−0.19	−0.07	−0.16	−0.57 *	−0.81 **

Note: The phenolic profiles are total phenolic contents (TPC), total flavonoid contents (TFC), total ferulic acid (TFA), total chlorogenic acid (TCA), total gallic acid (TGA), total vanallic acid (TVA), total syringate (TS), total 7-hydroxycoumarin (T7H), total quercetin (TQ), total quercetin 3-rhamnoside (TQ3). Total = free + bound. * and ** mean correlation is significant at the 0.05 and 0.01 levels, respectively (2-tailed).

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
