# Peer review of "Phytochemical Profiles and Cellular Antioxidant Activities in Chestnut (Castanea mollissima BL.) Kernels of Five Different Cultivars"

_molecules, 2020, doi:10.3390/molecules25010178_

Round 1

Reviewer 1 Report

This manuscript describes the phytochemical profiles and antioxidant activities of five different Chinese chestnut cultivars, which can add useful data to the literature.

In the abstract, use the full name of the chestnut cultivar.

Line 31-32: Include the reference.

It would be good to explain about other nutrients (e.g. vitamin E, C, carotenoids, etc.) in chestnut kernels to support that chestnuts are healthy food.

Line 31: Castanea crenata is a species originally native to Korea and Japan.

Line 33: Ref 1 doesn’t demonstrate the beneficial effect in human kidney.

Line 34: Delete “supplement”.

In section 2.1, describe the cultivar’s abbreviations when they are first introduced in the context.

Also, include the cultivar’s abbreviations in each figure and table.

Table 1 and 2: When ND is used, please include the limit of detection in the footnote.

Line 188: Delete  “(“.

Line 201- 202: Check the grammar.

Line 207: It’s not proper to use “good”. Use high or low.

Line 216: Include the growing condition for chestnut trees in orchard briefly.

Line 229: Describe the centrifugation condition.

Line 250: external curve or internal curve?

Line 264: Include the full name of DCFH-DA.

Provide the HPLC chromatogram of chestnut phenols and flavonoids.

Author Response

Reviewer 1

In the abstract, use the full name of the chestnut cultivar.

Response: Thank you for your suggestions. They have been added on the manuscript.

Line 31-32: Include the reference.

Response: Thank you for your suggestions. It was added.

It would be good to explain about other nutrients (e.g. vitamin E, C, carotenoids, etc.) in chestnut kernels to support that chestnuts are healthy food.

Response: Thank you for your suggestions. It was revised.

Line 31: Castanea crenata is a species originally native to Korea and Japan.

Response: Thank you for pointing it out. It has been corrected.

Line 33: Ref 1 doesn’t demonstrate the beneficial effect in human kidney.

Response: Thank you for pointing out. It was revised.

Line 34: Delete “supplement”.

Response: Thank you for pointing it out. It has been corrected.

In section 2.1, describe the cultivar’s abbreviations when they are first introduced in the context. Also, include the cultivar’s abbreviations in each figure and table.

Response: Thank you for the suggestion. The introduction to these five cultivars has been added.

Table 1 and 2: When ND is used, please include the limit of detection in the footnote

Response: Thank you for the suggestion. As table 1, all the standards concentration ranged from 1 μg/L to 10 μg/L as injection 10 μL each time in this study, if the peak of compounds was lower than the standard curve too much, it would be hard to area integral and calculation. As table 2, for cellular antioxidant activity assay, we used quercetin as standard to calculate the activities of extracts. Quercetin concentration was ranged from 1 to 10 μM. If the extracts were lower than standard curve too much. It would be hard to calculation. We defined the compounds or extracts which were lower than standard curves as “not detection” in this study.

Line 188: Delete “(“.

Response: thank you for pointing it out. It has been corrected.

Line 201- 202: Check the grammar.

Response: thank you for pointing it out. It has been corrected.

Line 207: It’s not proper to use “good”. Use high or low.

Response: Thanks for pointing it out. It has been corrected.

Line 216: Include the growing condition for chestnut trees in orchard briefly.

Response: It has been revised.

Line 229: Describe the centrifugation condition.

R: thank you for the notification. Centrifugation condition has been added.

Line 250: external curve or internal curve?

R: thank you for the notification. External curve has been added.

Line 264: Include the full name of DCFH-DA.

R: thank you for the suggestion. It has been added.

Provide the HPLC chromatogram of chestnut phenols and flavonoids.

Response: Thank you very much for your suggestion. The HPLC chromatograms of all the extracts as following:

Reviewer 2 Report

The manuscript “Phytochemical Profiles and Cellular Antioxidant Activities in Chestnut (Castanea mollissima BL.) Kernels of Five Different Cultivars” investigated the free and bound phenolic contents, flavonoid contents as well as phytochemical profiles in five different cultivars of chestnut kernels; moreover, total and cellular antioxidant activities were evaluated. The manuscript is interesting and original, the writing style and the use of English are accurate. The manuscript is well organized, conclusions are clear. However, the experimental design and the references can be improved. I have no specific scientific comment but it is necessary to correct the manuscript in few points before being considered for publication:

- please, check all the abbreviations used in the text. The abbreviations have to be defined in parentheses the first time they appear in the text.

- line 37. I suggest to underline the importance of natural bioactive compounds in human health. For this reason I suggest to read and cite more recent references, such as DOI: 10.1002/jsfa.7216, DOI: 10.1016/j.phrs.2017.12.020.

- in my opinion, considering that one of the main goal of this study is the determination of total antioxidant activities of these different extracts, it could be useful to perform other assay in this sense, such as TEAC or DPPH assay, which are commonly performed to test this parameter. Provide these data if it is possible.

Author Response

Reviewer 2

please, check all the abbreviations used in the text. The abbreviations have to be defined in parentheses the first time they appear in the text.

Response: thank you for the suggestion. They have been checked and full names were added.

line 37. I suggest to underline the importance of natural bioactive compounds in human health. For this reason I suggest to read and cite more recent references, such as DOI: 10.1002/jsfa.7216, DOI: 10.1016/j.phrs.2017.12.020.

Response: Thanks for suggestion. It has been added.

in my opinion, considering that one of the main goal of this study is the determination of total antioxidant activities of these different extracts, it could be useful to perform other assay in this sense, such as TEAC or DPPH assay, which are commonly performed to test this parameter. Provide these data if it is possible.

Response: Thank you for your advice. ORAC is a general method for total antioxidant activity assay from AOAC. This method has been listed as international standard for antioxidant activity analysis in many countries. There are pretty much research demonstrating good correlation among these antioxidant assays.